# Effects of Low-Phosphorus Diets Supplemented with Phytase on the Production Performance, Phosphorus-Calcium Metabolism, and Bone Metabolism of Aged Hy-Line Brown Laying Hens

**DOI:** 10.3390/ani13061042

**Published:** 2023-03-13

**Authors:** Yuechang Ren, Yaping Liu, Kexin Jiang, Linkui Li, Ning Jiao, Zhengqi Zhu, Kaiying Zhang, Shuzhen Jiang, Weiren Yang, Yang Li

**Affiliations:** 1Department of Animal Sciences and Technology, Shandong Agricultural University, Daizong Street 61#, Tai’an 271018, China; 2School of Food and Pharmaceutical Sciences, Ningbo University, Fenghua Road 818#, Ningbo 315832, China; 3Laiyang Animal Husbandry and Veterinary Station, Park Yu Road 888#, Laiyang 265200, China

**Keywords:** aged laying hens, low-phosphorus diet, phytase, phosphate rock, bone metabolism

## Abstract

**Simple Summary:**

In order to reduce feed costs and avoid environmental harm, laying hens should consume a limited amount of inorganic phosphorus in their diet. Phosphorus plays a vital role in bone and eggshell formation. In recent years, the rearing cycle of laying hens has lengthened. Skeletal problems in aged laying hens (as well as fractures during transport of culled hens) are considered to be the main threats to the extended rearing of laying hens. Therefore, this study was conducted to evaluate the effects of phytase supplementation in low-phosphorus diets on the production performance, phosphorus–calcium metabolism, and bone metabolism in laying hens from 69 to 78 weeks of age. The results indicated that a reduction in non-phytate phosphorus supplementation to 0.15% (dietary non-phytate phosphorus level = 0.27%) with phytase inclusion did not have an adverse effect on the production performance or bone health of laying hens from 69 to 78 weeks of age, which might be attributed to renal phosphorus reabsorption and bone resorption. These findings could support the application of low-phosphorus diets in the poultry industry.

**Abstract:**

This study was conducted to evaluate the effects of phytase supplementation in low-phosphorus diets on the production performance, phosphorus–calcium metabolism, and bone metabolism in laying hens from 69 to 78 weeks of age. Hy-Line Brown laying hens (*n* = 1350) were assigned randomly to six treatments with five replicates of 45 birds. A corn–soybean meal-based diet with no inorganic phosphates was formulated to contain 0.12% non-phytate phosphorus (NPP) and 1470 FTU/kg phytase (Released phytate phosphorus content ≥ 0.1%). Inorganic phosphorus (dicalcium phosphate) was supplemented into the basal diet to construct five test diets (level of NPP supplementation = 0.10%, 0.15%, 0.20%, 0.25%, and 0.30%). The level of calcium carbonate was adjusted to ensure that all six experimental diets contained the same calcium percentage (3.81%). The feeding trial lasted 10 weeks (hens from 69 to 78 weeks of age). Upon supplementation with phytase (1470 FTU/kg), supplemental inorganic phosphates (dicalcium phosphate) had no significant effects (*p* > 0.05) on the production performance or egg quality. Significant differences in serum levels of calcium, phosphorus, copper, iron, zinc, or manganese were not detected across treatments (*p* > 0.05). Hens fed NPP (0.15%, 0.20%, 0.25%, and 0.30%) had higher levels (*p* < 0.0001) of tibial ash, calcium, and phosphorus than those not fed inorganic phosphates. The tibial breaking strength of the group without inorganic phosphates was significantly lower than that of the other groups (*p* < 0.01). Dietary supplementation with inorganic phosphates had no effect (*p* > 0.05) on serum levels of calcitonin (CT) and 1,25-dihydroxy-vitamin D3 (1,25-(OH)2D3). Hens that did not receive supplementation with inorganic phosphates had higher serum levels of parathyroid hormone (PTH), osteoprotegerin (OPG), type-I collagen c-telopeptide (CTX-I), and tartrate-resistant acid phosphatase 5b (TRACP-5b) compared with those in the other groups (*p* < 0.01). Serum levels of CTX-I and TRACP-5b were significantly lower in the NPP-supplementation groups of 0.25% and 0.30% than in the 0.10% NPP-supplementation group (*p* < 0.01). Dietary supplementation with inorganic phosphates had no effect (*p* > 0.05) on serum levels of bone-alkaline phosphatase (BAP), osteocalcin (OCN), or osteopontin (OPN). Hens not fed inorganic phosphate had the highest renal expression of phosphorus transporter type IIa Na/Pi cotransporter (NaPi-Ⅱa). Renal expression of NaPi-Ⅱa was increased significantly in NPP-supplementation groups of 0.10–0.20% compared with that in NPP-supplementation groups of 0.25% and 0.30% (*p* < 0.0001). The results indicated that a reduction in NPP supplementation to 0.15% (dietary NPP level = 0.27%) with phytase inclusion did not have an adverse effect on the production performance or bone health of laying hens from 69 to 78 weeks of age, which might be attributed to renal phosphorus reabsorption and bone resorption. These findings could support the application of low-phosphorus diets in the poultry industry.

## 1. Introduction

Phosphorus is an essential nutrient for animals and plays a vital role in bone and eggshell formation. Corn meals and soybean meals have a relatively high content of phosphorus, but most phosphorus is present as phytic acid [1]. Mgonogastric animals lack sufficient endogenous phytase to hydrolyze phytate complexes in grains and, therefore, have poor utilization of phytate phosphorus [2]. Hence, supplementation with inorganic phosphates is necessary to support the normal growth of monogastric animals [3]. However, inorganic phosphate input in animal diets is directly related to the conservation of phosphate rocks and environmental concern [4]. In order to reduce feed costs and avoid environmental harm, laying hens should consume a limited amount of inorganic phosphorus in their diet. In recent decades, phytase has been widely applied in the feed industry due to its ability to break down phytate-phosphate from plants and reduce the animals’ requirement for inorganic phosphates [5]. Numerous studies have shown that phytase dosages > 1000 FTU/kg could further improve the utilization of phytate-phosphate and reduce the amount of inorganic phosphates, even without extra supplementation [6,7,8,9]. Wang et al. [10] reported that laying performance could be maintained if phytase at 360 FTU/kg was supplemented into a basal diet (NPP = 0.16%) with supplementation of <0.10% phosphorus. Some studies showed that if phytase was supplemented into a diet, a reduction in the NPP level to 0.25% did not have an adverse effect on bone mineralization in laying hens [11]. Similarly, eggshell quality was not significantly enhanced if the dietary NPP level was increased to 0.40% or 0.30% [12]. Based on our understanding of the current literature, low-phosphorus diets are feasible for laying hens. If so, reducing inorganic phosphates would be an environmentally friendly and cost-effective feed-processing strategy.

In recent years, the rearing cycle of laying hens has lengthened. Skeletal problems in aged laying hens (as well as fractures during transport of culled hens) are considered to be the main threats to the extended rearing of laying hens. The bone of laying hens consists of cortical, trabecular, and medullary bone tissues. Medullary bone is a nonstructural tissue formed after sexual maturity that stores mineral elements such as calcium and phosphorus to support eggshell formation [13,14]. Bone is a dynamic organ that maintains a dynamic balance through osteoblasts and osteoclasts [15]. The levels of osteoprotegerin (bone formation marker) and tartrate-resistant acid phosphatase 5b (bone resorption marker) in serum have been used to assess bone metabolism in hens [16,17]. Serum calcium, phosphorus, and 1,25-(OH)2D3 concentrations have a vital role in bone mineralization, thus their levels can reflect bone health [17,18,19]. However, relatively few studies have focused on the effect of the level of NPP supplementation on bone metabolism in aged laying hens when phytase was added to basal diets. We hypothesized that low-phosphorus diets would have no adverse effects on production performance or bone metabolism if aged laying hens were fed a corn–soybean meal-based diet containing phytase at 1470 FTU/kg. In the present study, dicalcium phosphate was supplemented into the basal diet to create six phytase-containing experimental diets (level of NPP supplementation = 0, 0.10%, 0.15%, 0.20%, 0.25%, and 0.30%). Production performance, egg quality, phosphorus–calcium metabolism, and bone metabolism were evaluated in Hy-Line Brown laying hens from 69 weeks to 78 weeks of age.

## 2. Materials and Methods

### 2.1. Experimental Design and Feeding Management

The study protocol was approved (S20180058) by the Animal Protection and Use Committee of the College of Animal Science and Technology within Shandong Agricultural University (Taian, China). Hy-Line Brown laying hens (*n* = 1350; live weight (LW) = 1.98 kg ± 0.16) were assigned randomly to six treatments with five replicates of 45 birds. The hens were housed in cages with raised wire floors (depth × width × height = 47 × 36 × 105 cm). There were three hens in each cage. The composition of the corn-soybean meal-based diet lacking inorganic phosphate is shown in Table 1. Briefly, the basal diet contained crude protein (15.79%), calcium (3.81%), NPP (0.12%), and phytase (1470 FTU/kg). All hens are fed the same basal diet. Inorganic phosphate (dicalcium phosphate) was supplemented into the basal diet to construct five test diets (level of NPP supplementation = 0.10%, 0.15%, 0.20%, 0.25%, and 0.30%). The level of calcium carbonate was adjusted to ensure that all six experimental diets contained an identical level of calcium (3.81%). Phytase (≥50,000 FTU/g; measured value: 50,511 FTU/g; commercial name: Habio phytase; 6-phytase produced by *Escherichia coli*) was purchased from Jinan Bestzyme Bio-engineering (Jinan, China). The mineral levels and phytase activity of experimental diets are shown in Table 2.

Prior to the feeding trial (from 67 to 68 weeks of age), all the hens were fed the same commercial diet (NPP = 0.32%). Then, the hens were randomly assigned to the six experimental diets. The feeding trial lasted 10 weeks (hens from 69 to 78 weeks of age). Feed and water were provided ad libitum. Hens were exposed to a 16 h/8 h light-dark photoperiod in the egg farm within Shandong He-Mei-Hua Agricultural and Animal Technology (Jinan, China). The temperature in the laying room was maintained between 17.0 °C and 21.0 °C.

### 2.2. Production Performance and Egg Quality

The total weight and number of eggs were recorded daily. Feed consumption was measured weekly to calculate daily feed intake and the feed-to-egg mass ratio. At 78 weeks of age, 10 eggs per replicate (50 eggs per treatment) were selected randomly to determine egg quality. Albumen height, yolk color, and Haugh units were determined using an egg multi-tester (EMT-5200; Robotmation, Tokyo, Japan). Eggshell strength was measured using a strength gauge (EFG-0503; Robotmation).

### 2.3. Serum Parameters

At 78 weeks of age, two hens per replicate were randomly selected to undergo bloodletting via a wing vein between 7 am and 11:30 am (10 mL per hen). Blood samples were centrifuged (3500 rpm, 10 min, room temperature; GL-18MC; Hunan Hu Kang Centrifuges, Changsha, China). Serum samples were collected using centrifugal tubes and stored at −80 °C. Serum levels of alkaline phosphatase (ALP), calcium, and phosphorus were analyzed using a fully automatic biochemical analyzer (L-7020; Hitachi, Tokyo, Japan).

### 2.4. Determination of Levels of Copper, Iron, Manganese, and Zinc in Serum

Serum samples were analyzed using a flame atomic absorption spectrometer (Z-8200; Hitachi) according to the method described by Andreani and colleagues [20]. Element concentrations are reported as pmol/L.

### 2.5. Determination of Serum Calcium and Phosphorus Metabolism-Related Hormones

Calcium and phosphorus metabolism-related hormones, including CT, PTH, and 1,25-(OH)2D3, were determined with the relevant enzyme-linked immunosorbent assay (ELISA) kits (Jiangsu Meimian Industrial, Yancheng, China) according to manufacturer instructions.

### 2.6. Tibia Quality

Immediately after being bled, hens were euthanized by cervical dislocation. Both femurs were removed for compositional analysis and determination of strength. We wished to determine the breaking strength of the right tibia, so it was subjected to a three-point bending test using a microcomputer-controlled electronic universal mechanical test machine (YAW-5000F; Jinan Pilot Gold Group, Jinan, China). The preload speed of the tester was set to 2 mm/min and the support distance was set to 40 mm. The left tibia was degreased with petroleum ether for 8 h and then dried in an oven at 105 °C. Skimmed tibia samples were turned to ash in a muffle furnace (600 °C, 8 h). Then, we measured the levels of calcium (using the EDTA titration method) and phosphorus (using a colorimetric method based on ammonium metavanadate). Then, the contention of defatted-and-oven-dried tibial ash, calcium, and phosphorus were calculated accordingly.

### 2.7. Determination of Serum Bone Turnover Markers

Serum bone formation markers, including BAP, OCN, and OPG, were determined with the relevant ELISA kits (Jiangsu Meimian Industrial) according to manufacturer instructions.

Serum bone resorption markers, including CTX-I, OPN, and TRACP-5b, were determined with the relevant ELISA kits (Jiangsu Meimian Industrial) according to manufacturer instructions.

### 2.8. Real-Time Reverse Transcription-Quantitative Polymerase Chain Reaction (RT-qPCR)

After the hens were euthanized, the intact kidneys were removed and cleaned with 70% alcohol. Kidney samples were stored at −80 °C using liquid nitrogen to determine the expression of phosphorus transporters type IIa Na/Pi cotransporter (NaPi-Ⅱa). Total RNA of the kidney was extracted using TRIzol^®^ Reagent (Invitrogen, Carlsbad, CA, USA). Samples of total RNA were subjected to reverse transcription using PrimeScript™ RT Master Mix (TaKaRa Biotechnology, Shiga, Japan) for complementary DNA synthesis. Then, RT-qPCR was carried out using the SYBR Premix Ex Taq kit (TaKaRa Biotechnology) in a real-time PCR system (ABI7500; Applied Biosystems, Foster City, CA, USA). β-actin was the internal reference gene. The primer sequences are shown in Table 2. The amplification procedure was: 95 °C for 10 min (95 °C for 15 s, 60 °C for 30 s) × 40 cycles, and 95 °C for 15 s. Relative mRNA expression was calculated using the 2^−△△ct^ method. The primer sequence and product size are shown in Table 3.

### 2.9. Statistical Analyses

Statistical analyses were carried out using SAS 9.2 (SAS Institute, Cary, NC, USA). One-way ANOVA was conducted to detect differences among groups. Duncan’s test was applied if significance was detected in the one-way ANOVA. The statistical unit was each replicate (*n* = 5) for production-performance parameters; each egg (*n* = 50; 10 eggs per replicate were selected randomly) for egg-quality parameters; and each hen (*n* = 10; two hens per replicate were selected randomly) for measurements in samples of serum, tibia, and kidney. Data are the mean ± standard error. Significance was considered at *p* < 0.05.

## 3. Results

### 3.1. Production Performance and Egg Quality

Significant differences were not observed (*p* > 0.05) among treatments for production performance, including day laying rate, average egg weight, feed-to-egg mass ratio, broken egg rate, and feed intake (Figure 1). Supplementation with inorganic phosphate had no effect on egg quality as measured by albumen height, eggshell strength, yolk color, and Haugh units (Figure 1).

### 3.2. Serum Parameters, Serum Micronutrients, Serum Calcium and Phosphorus Metabolism-Related Hormones, and Tibia Quality

Significant differences were not observed (*p* > 0.05) in serum levels of calcium, phosphorus, ALP, copper, iron, zinc, or manganese across treatments (Table 4). Dietary supplementation with inorganic phosphates had no effect (*p* > 0.05) on serum levels of CT and 1,25-(OH)2D3. Hens not supplemented with inorganic phosphates had higher (*p* < 0.01) serum levels of PTH than those in the other groups (Figure 2).

Hens fed NPP (0.15%, 0.20%, 0.25%, or 0.30%) had higher levels (*p* < 0.0001) of tibial ash, calcium, and phosphorus compared with those not fed inorganic phosphate (Table 5). The tibial breaking strength of hens in the group not supplemented with inorganic phosphate was significantly lower than that of other treatment groups (*p* < 0.01). There was no significant difference in tibial calcium content between groups supplemented with NPP at 0.10%, 0.15%, or 0.20% (*p* > 0.05).

### 3.3. Serum Bone Turnover Markers and Renal Expression of NaPi-Ⅱa

Dietary supplementation with inorganic phosphate had no effect on serum levels of calcitonin, 1,25-(OH)2D3, BAP, OCN, or OPN. Hens not supplemented with inorganic phosphate had higher serum levels of PTH, OPN, CTX-I, and TRACP-5b than those in the other groups (*p* < 0.01). (Figure 2, Figure 3 and Figure 4). Serum levels of CTX-I and TRACP-5b were significantly lower in the NPP-supplementation groups of 0.25% and 0.30% than those in the 0.10% NPP-supplementation group (*p* < 0.01) (Figure 4). Hens not fed inorganic phosphate had the highest renal expression of NaPi-Ⅱa. Renal expression of NaPi-Ⅱa was increased significantly by NPP supplementation of 0.10% and 0.20% compared with that induced by NPP supplementation of 0.25% or 0.30% (*p* < 0.0001) (Figure 5).

## 4. Discussion

Some related studies reported that supplementing microbial phytase to a basal diet without inorganic phosphate could maintain laying rate and egg weight satisfactorily [21,22,23]. Similarly, Ahmadi and Rodehutscord [24] found that 0.14% NPP in the basal diet is adequate for laying hens fed with 400 FTU/kg phytase. The current study showed that production performance and egg quality were not significantly different among dietary treatments if phytase at 1470 FTU/kg was supplemented into a diet. Laying hens can adapt physiologically to a low-phosphorus diet and maintain normal production performance and egg quality depending on the degree of phosphorus deficiency, production process, and phytase dose [11,25,26]. The results did not reveal adverse effects on production performance or egg quality, so supplementing large amounts of inorganic phosphates under current conditions would waste significant phosphorus resources. For premix companies, low levels of NPP coupled with phytase should be a good option for cost reduction and efficiency. To draw solid conclusions, long-term field studies are needed to determine the optimal level of NPP supplementation in feeds.

Tibial quality is considered an essential factor for evaluating phosphorus uptake in poultry. An excess or deficiency of phosphorus in a diet can result in bone disease in poultry [27,28]. Carlos and Edwards Jr. [21] reported a significant improvement in bone quality if phytase was added to a diet containing a low NPP level. Jing et al. [29] indicated that tibial weight, tibial ash, calcium percentage, and phosphorus percentage were not different for Lohmann White laying hens from 22 weeks to 34 weeks of age fed various levels of NPP if phytase at 1000 FTU/kg was added to the diet. Cheng et al. [30] reported that 0.12% NPP in a basal diet (without inorganic phosphate) did not cause significant changes in the tibiae of Hy-Line Brown laying hens from 29 weeks to 40 weeks of age with supplementation of phytase at 2000 FTU/kg. In general, for laying hens, there is no need to add inorganic phosphate to a diet containing phytase [9,24,30]; our data did not seem to confirm this view. Tibial breaking strength, tibial ash, and phosphorus content were significantly lower in the group without inorganic phosphate (dietary NPP level = 0.12%) compared with the group with NPP supplementation of 0.15–0.30% (dietary NPP level = 0.27–0.42%). The results obtained from different studies are not entirely consistent, which may be related to breed and day-age differences as well as the breeding environment. Some studies have shown the age of the animal to be an important factor affecting phosphorus utilization, and aged hens may be more sensitive to a dietary deficiency of phosphorus [31].

Viveros et al. [32] reported that the dietary NPP level was linearly correlated with the plasma phosphorus level in poultry. In the current study, significant differences in serum levels of calcium, phosphorus, or ALP were not observed in the NPP-supplementation groups of 0.10–0.30% (dietary NPP level = 0.22–0.42%) compared with the group not given inorganic phosphates (dietary NPP level = 0.12%). Similarly, Boorman and Gunaratne [33] reported that serum levels of calcium and phosphorus in laying hens from 25 to 36 weeks of age were not affected by diets with NPP levels of 0.16%, 0.21%, 0.31%, or 0.39%. The lack of significant differences in dietary NPP levels on serum levels of calcitonin, ALP, phosphorus, or calcium might be due to the regulation of environmental homeostasis in animals [34]. PTH, CT, and 1,25-(OH)2D3 in the serum are key indicators for phosphate–calcium metabolism balance [34,35]. In the current study, hens not supplemented with inorganic phosphates had higher serum levels of PTH than those in the other groups. PTH promotes the reabsorption of phosphorus by the kidney and maintains the body’s phosphorus levels at a normal level [36].

OPG is an irreducible soluble receptor secreted by osteoblasts, which could block the formation of osteoclasts, thereby protecting bones [37]. Hens fed diets without phosphates addition (dietary NPP level = 0.12%) had higher serum levels of OPG than those in other groups. The serum analyzed for the bone turnover markers was collected between 8:30 am and noon, which is around the time when most hens lay eggs [38]. At this time, the calcium needed by the hen to form the eggshell is mainly taken up from the diet and not from the bones [39]. The serum level of OPG was increased significantly in the group not fed phosphate, indicating that the speed of bone transformation was accelerated due to the acceleration of bone resorption, which resulted in bone loss [40]. This bone loss can stimulate and exacerbate the compensatory proliferation of osteoblasts to inhibit abnormal bone resorption [41,42]. TRACP-5b is derived from osteoblasts and is involved in bone degradation during bone resorption. The serum level of TRACP-5b is not readily affected by diseases, and it is a specific indicator for identifying bone loss [43,44]. CTX-Ⅰ is the main bone organic collagen crosslink. Under pathological stimulation, osteoclast activity is enhanced, which results in the degradation of large amounts of type-I collagen to CTX-Ⅰ in the blood. The physiological variation of CTX-Ⅰ in serum is small and could reflect bone resorption specifically [45]. The increase in serum levels of CTX-I and TRACP-5b indicated that low phosphorus intake caused hens to mobilize calcium and phosphorus in bone to support eggshell formation. Laying hens from 69 weeks to 78 weeks of age might maintain egg production by increasing bone resorption. Genetic selection over the years has improved the adaptability of laying hens to bone loss [46].

Huber et al. [47] reported that the capacity of the kidney for phosphorus transport is important for maintaining phosphorus homeostasis in laying hens. Hens not fed inorganic phosphates (dietary NPP level = 0.12%) had the highest renal expression of NaPi-Ⅱa. The higher NaPi-Ⅱa expression in NPP-supplementation groups of 0.10–0.20% (dietary NPP level = 0.22–0.32%) compared with that in NPP-supplementation groups of 0.25% and 0.30% (dietary NPP level = 0.37% and 0.42%) indicated that a low-phosphorus diet may increase renal phosphorus reabsorption. This result is similar to the data of Li et al. [48], who reported that renal NaPi-IIa expression in hens fed 0.15% available phosphorus (AP) was significantly higher than that in hens fed AP at 0.41% or 0.82%. Lotscher and coworkers [49] reported that renal NaPi-IIa expression was reduced significantly in rats after continuous feeding of high-phosphorus diets for 7 days. Antibody against NaPi-Ⅱa was not purchased, so protein expression of NaPi-Ⅱa in the kidney was not measured in our study. Hence, the effect of a low-phosphorus diet on phosphorus balance remains to be elucidated.

## 5. Conclusions

A reduction in the NPP supplementation level to 0.15% (dietary NPP level = 0.27%) with the inclusion of phytase did not have an adverse effect on the production performance or bone health of laying hens from 69 weeks to 78 weeks of age. These phenomena could be attributed to renal phosphorus reabsorption and bone resorption. These findings support the application of low-phosphorus diets in the poultry industry.

## Figures and Tables

**Figure 1 animals-13-01042-f001:**
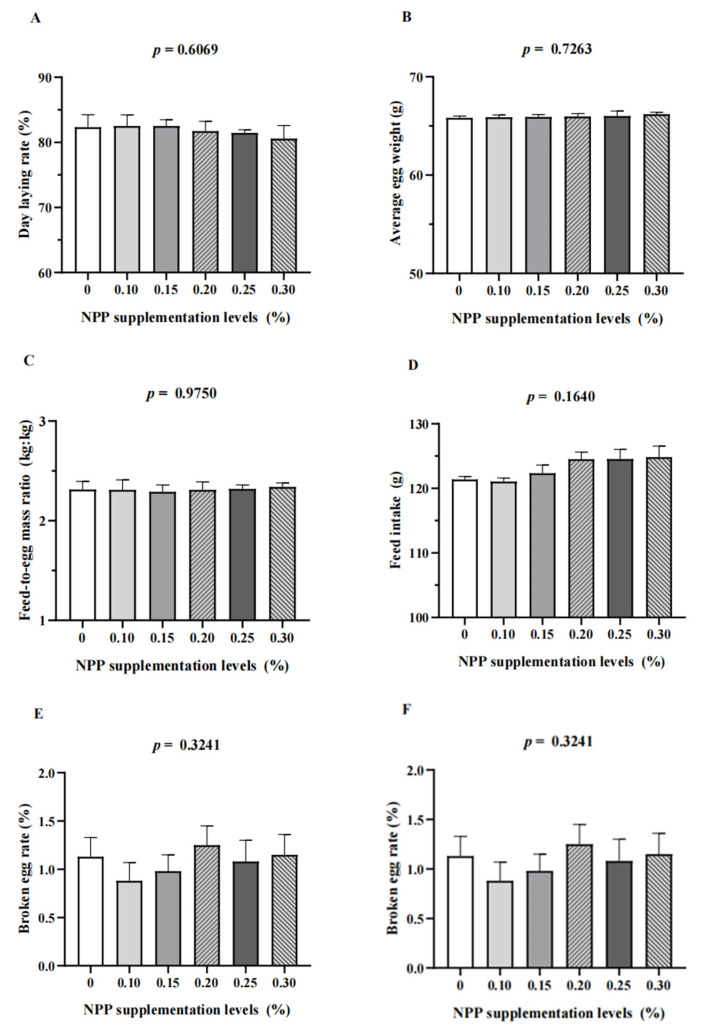
Effect of the level of dietary NPP supplementation on the production performance and egg quality of Hy-Line Brown laying hens (69–78 weeks of age) fed a phytase-containing diet. Production performance (**A**–**E**); egg quality (**F**–**I**). All data were subjected to linear analyses, but significance was not observed (*p* > 0.05). The statistical unit was each replicate (*n* = 5) for the production performance and was each egg (*n* = 50; 10 eggs per replicate were selected randomly) for egg-quality parameters.

**Figure 2 animals-13-01042-f002:**
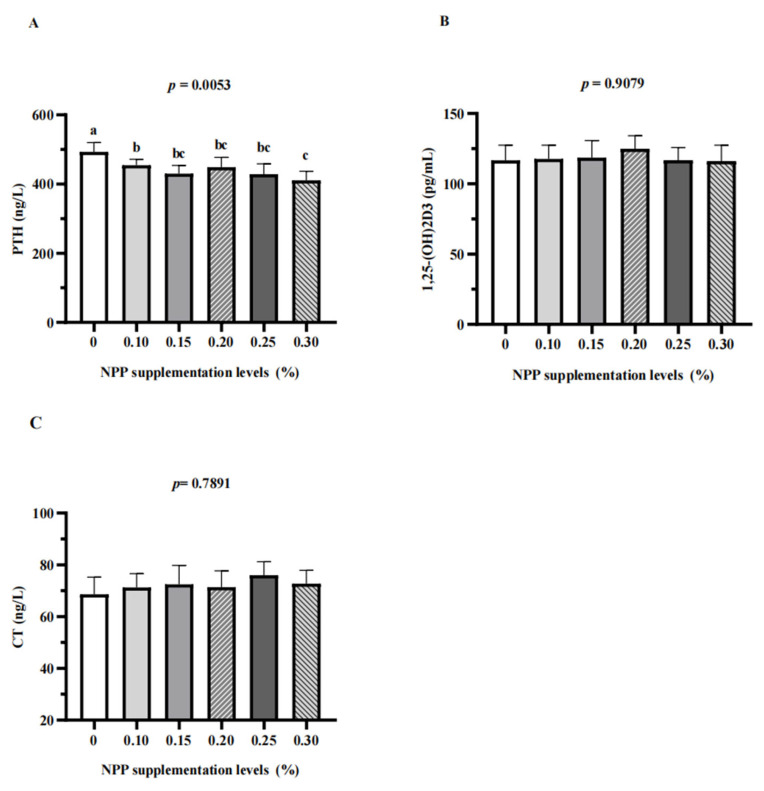
Effect of the level of dietary NPP supplementation on the serum calcium and phosphorus metabolism-related hormones of Hy-Line Brown laying hens (69–78 weeks of age) fed a phytase-containing diet. PTH = parathyroid hormone (**A**); 1,25-(OH)2D3 = 1,25-dihydroxy-vitamin D3 (**B**); CT = calcitonin (**C**). ^a–c^ Mean values within one row lacking a common superscript differ (*p* < 0.01). The statistical unit was each hen (*n* = 10; at the end of feeding, two hens per replicate were selected randomly for sample collection).

**Figure 3 animals-13-01042-f003:**
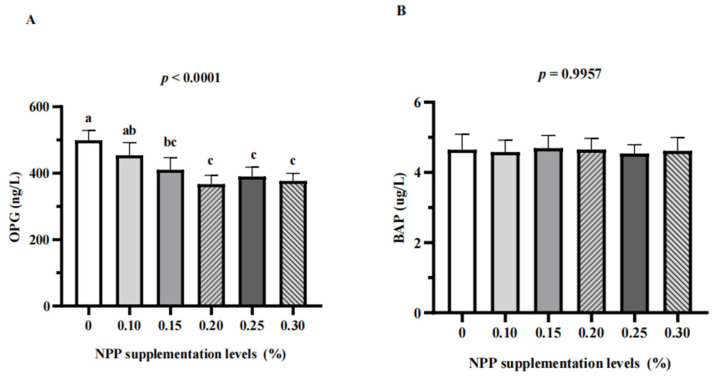
Effect of the level of dietary NPP supplementation on the serum bone formation markers of Hy-Line Brown laying hens (69–78 weeks of age) fed a phytase-containing diet. OPG = osteoprotegerin (**A**); BAP = bone alkaline phosphatase (**B**); OCN = osteocalcin (**C**). ^a–c^ Mean values within one row lacking a common superscript differ (*p* < 0.01). The statistical unit was each hen (*n* = 10; at the end of feeding, two hens per replicate were selected randomly for sample collection).

**Figure 4 animals-13-01042-f004:**
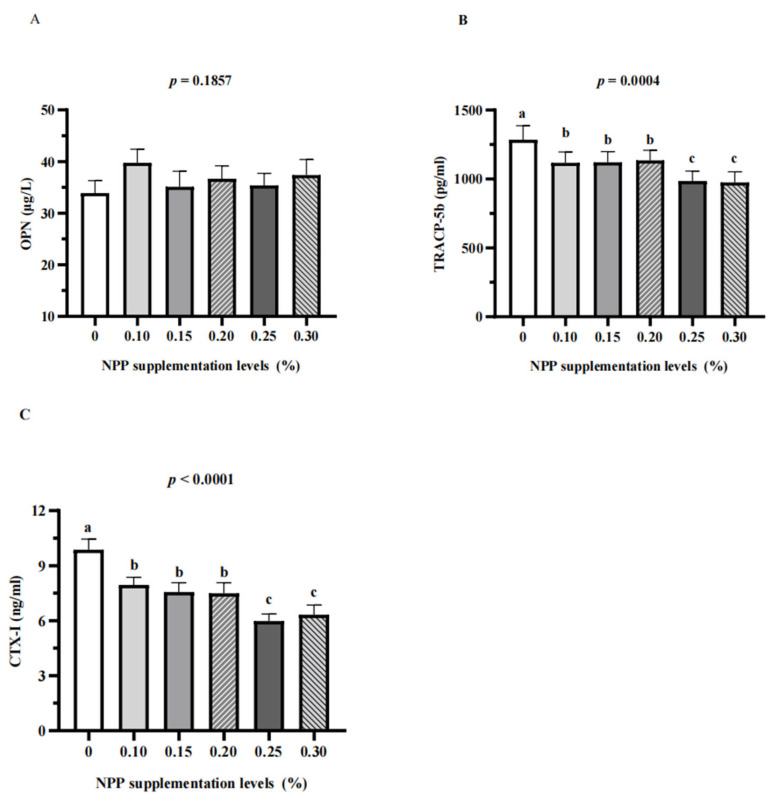
Effect of the level of dietary NPP supplementation on the serum bone resorption markers of Hy-Line Brown laying hens (69–78 weeks of age) fed a phytase-containing diet. OPN = osteopontin (**A**); TRACP-5b = tartrate-resistant acid phosphatase 5b (**B**); CTX-I = C-telopeptide of type-I collagen (**C**). ^a–c^ Mean values within one row lacking a common superscript differ (*p* < 0.01). The statistical unit was each hen (*n* = 10; at the end of feeding, two hens per replicate were selected randomly for sample collection).

**Figure 5 animals-13-01042-f005:**
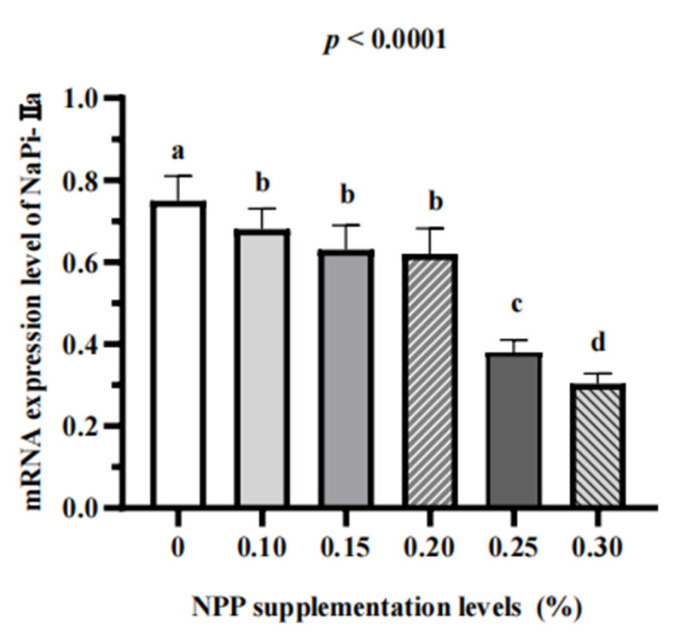
Effect of the level of dietary NPP supplementation on the expression of phosphorus transporters type IIa Na/Pi cotransporter (NaPi-Ⅱa) in the kidneys of Hy-Line Brown laying hens (69–78 weeks of age) fed a phytase-containing diet. ^a–d^ Mean values within one row lacking a common superscript differ (*p* < 0.01). The statistical unit was each hen (*n* = 10; at the end of feeding, two laying hens per replicate were selected randomly for sample collection). Relative mRNA expression was calculated for the housekeeping gene β-actin using the 2^−ΔΔCt^ method.

**Table 1 animals-13-01042-t001:** Composition and nutrient contents of the basal diet.

Item	% (Unless Stated Otherwise)
Corn	60.5
Soybean meal, 46%	22
Soybean oil	1
Calcium carbonate	8.5
Palmmeal	5
Dicalcium phosphate	-
Premix ^1^	3
Total	100
Nutrient contents	
Metabolic energy (Kcal/kg)	2537
Crude protein	15.79
Phytase (FTU/kg)	1470
Calcium	3.81
Total phosphorus (analyzed)	0.34
NPP	0.12
Lysine	0.81
Methionine	0.42

^1^ The premix provided the following per kilogram of diet: vitamin A, 8100 IU; vitamin D3, 2700 IU; vitamin E, 24 mg; vitamin K3, 3.0 mg; vitamin B1, 2.4 mg; vitamin B2, 6 mg; vitamin B6, 3 mg; Niacin, 10 mg; folic acid, 1.2 mg; pantothenic, 7.8 mg; VB12, 0.003 mg; copper (from CuSO_4.5_H_2_O), 10.5 mg; manganese (from MnSO_4_.H_2_O), 84 mg; zinc (from ZnSO_4.6_H_2_O), 78 mg; iron (from FeSO_4.6_H_2_O), 54.0 mg; selenium (from Na_2_SeO_3.5_H_2_O), 0.5 mg; iodine (from KT), 0.45 mg; lysine, 0.7 g; methionine, 1.77 g; NaCl, 3 g; phytase, 29.1 mg.

**Table 2 animals-13-01042-t002:** Mineral levels and phytase activity of experimental diets.

Treatment	Phytase (mg/kg)	Dietary Phytase Activity (FTU/kg)	Calcium %	NPP Supplementation Levels %	Dietary NPP Levels %
1	29.1	1470	3.81	0	0.12
2	29.1	1470	3.81	0.10	0.22
3	29.1	1470	3.81	0.15	0.27
4	29.1	1470	3.81	0.20	0.32
5	29.1	1470	3.81	0.25	0.37
6	29.1	1470	3.81	0.30	0.42

**Table 3 animals-13-01042-t003:** Primer sequences used for real-time RT-qPCR.

Target	Primer	Primer Sequence	Predicted Size (bp)	Annealing Temperature (°C)
Gene				
NaPi-Ⅱa	upstream	5′-TTGGCCAGCACCAGACTCAG-3′	118	60
	downstream	5′-GCAGTAGGCAAACGGGGTC-3′		60
β-actin	upstream	5′-CTGGGGCTCATCTGAAGGGT-3′	308	60
	downstream	5′-GGACGCTGGGATGATGTTCT-3′		60

**Table 4 animals-13-01042-t004:** Effect of levels of dietary NPP supplementation on the serum parameters and serum micronutrients of Hy-Line Brown laying hens (69–78 weeks of age) fed a phytase-containing diet ^1,2^.

	NPP Supplementation Levels (%)						
Items	0	0.1	0.15	0.20	0.25	0.30	*p*-value
Serum parameters							
Calcium (mmol/L)	4.36 ± 0.40	4.96 ± 0.36	4.52 ± 0.38	4.64 ± 0.43	4.63 ± 0.33	4.77 ± 0.42	0.1880
Phosphorus (mmol/L)	1.46 ± 0.18	1.52 ± 0.10	1.64 ± 0.17	1.50 ± 0.16	1.68 ± 0.17	1.66 ± 0.11	0.1309
ALP (U/L)	381.00 ± 27.00	407.00 ± 46.89	394.67 ± 50.54	340.00 ± 35.17	377.00 ± 41.07	423.67 ± 42.62	0.2445
Serum micronutrients							
Copper (pmol/L)	32.74 ± 2.83	34.30 ± 2.19	32.81 ± 2.81	33.55 ± 4.42	34.63 ± 2.34	32.72 ± 2.32	0.7700
Iron (pmol/L)	20.05 ± 1.05	20.25 ± 1.11	19.80 ± 2.27	19.55 ± 1.40	20.87 ± 2.67	20.86 ± 3.21	0.8436
Zinc (pmol/L)	60.38 ± 4.97	58.83 ± 5.34	60.24 ± 3.69	57.70 ± 4.68	63.58 ± 5.82	60.98 ± 6.78	0.5260
Manganese (pmol/L)	10.11 ± 1.08	9.71 ± 0.86	10.37 ± 0.69	9.82 ± 1.34	10.25 ± 0.55	10.58 ± 0.45	0.5383

NPP = non-phytate phoNPP = non-phytate phosphorus; ALP = alkaline phosphatase. ^1^ Data are the mean ± standard error of the mean. Phytase was used at 1470 FTU/kg in diets. ^2^ All data were subjected to linear analyses, but significance was not observed (*p* > 0.05).

**Table 5 animals-13-01042-t005:** Effect of the level of dietary NPP supplementation on tibial quality of Hy-Line Brown laying hens (69–78 weeks of age) fed a phytase-containing diet ^1,2,3^.

	NPP Supplementation Levels (%)						
Items	0	0.1	0.15	0.20	0.25	0.30	*p*-value
Breaking strength (N/mm²)	13.21 ^c^ ± 2.68	15.49 ^b^ ± 1.48	15.33 ^b^ ± 1.09	15.41 ^b^ ± 1.89	17.46 ^a^ ± 1.01	17.98 ^a^ ± 2.82	0.0061
Ash content (%)	58.56 ^d^ ± 0.41	59.83 ^bc^ ± 0.67	59.75b ^c^ ± 0.51	60.40 ^ab^ ± 0.42	60.61 ^a^ ± 0.41	61.02 ^a^ ± 0.46	<0.0001
Calcium (%)	19.62 ^d^ ± 0.32	20.16 ^cd^ ± 0.30	20.13 ^cd^ ± 0.39	20.50 ^bc^ ± 0.28	20.88 ^ab^ ± 0.39	21.15 ^a^ ± 0.29	<0.0001
Phosphorus (%)	6.91 ^d^ ± 0.65	7.62 ^c^ ± 0.39	7.80 ^bc^ ± 0.50	8.47 ^ab^ ± 0.47	8.80 ^a^ ± 0.52	9.03 ^a^ ± 0.75	<0.0001

NPP = non-phytate phosphorus. ^1 a–d^ Mean values with different superscripts within one row were significantly different (*p* < 0.05). ^2^ Data are the mean ± standard error of the mean. Phytase was used at 1470 FTU/kg in diets. ^3^ The statistical unit was each hen (*n* = 5; at the end of feeding, two laying hens per replicate was selected randomly for sample collection).

## Data Availability

Data supporting the reported results is contained within the article.

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
