# Peer review of "Effects of Low-Phosphorus Diets Supplemented with Phytase on the Production Performance, Phosphorus-Calcium Metabolism, and Bone Metabolism of Aged Hy-Line Brown Laying Hens"

_animals, 2023, doi:10.3390/ani13061042_

Round 1
Reviewer 1 Report
A well written paper. A few minor suggestions.
The main concern is the conclusion about low NPP diets with phytase. In the conclusions per se it states that low NPP +Phytase diets do not impact performance or bone health. But this statement is followed by the conclusion that these diets may cause skeletal issues. So- what is the conclusion for readers?
L17 ..environmental issues.
L27 Explain replication
L52 Conclusion at odds with previous sentence??
L98 Tables 2 and 3 seems out of place- should be in Results section.
Reviewer 2 Report
Comments to the Authors of manuscript number: animals-2223882 entitled “Effects of Low-phosphorus Diets Supplemented with Phytase on the Production Performance and Bone Metabolism of Aged Hy-Line Brown Laying Hens”.
There is too much substantial methodological mistakes. For this reason the paper is not suitable for publication.
1. L. 17- environmental what?
2. L18- abbr.
3. L 21 – there is no “the metabolism of bone resorption”.
4. L 27 – bieds?
5. L 83 – other hormones or factors are indicators of bone metabolism, and there are such studies performed on laying hens showing them
6. L 99 – what was real activity of phytase? Was it checked?
7. phosphorus was given on top? The composition of diet cannot exceed 100%
8. There should be presented all diets because of it is not known how all diets were formulated, and how this inorganic phosphorus was given.
9. the period of the study is only mentioned in abstracts, but in material and methods it is not explained why these ten weeks were chosen. Authors wrote that the laying period was extended. Are these 10 weeks? What is so important in these weeks that this time is studying? It should be explained. Other researches have to know
10. L 110- table 1 does not include phytase
11. the diet is formulated not properly
12. L 111- it is not presented
13. L 112- what information about the activity of phytase is given by producer?
14. L 115- total weight of what?
15. It is unclear how birds were located? There is no information how many birds were in one box, thus it is not known how feed consumption was calculated. And how the number of eggs were calculated. How many eggs can be from one bird? One bird could give less and another more, and if in the box a few birds were, it is false results.
16. the daily intake was calculated based on the weekly collected data about diet. It is false result
17. L 117- it is not know how many eggs is from one bird. Maybe, the predominate amount is from one bird.
18. It is unclear how long this diet was given.
19. There is no data about performance and diet before introduction of the diet tested.
20. there are no data about performance before the diet start.
21. there is only one time point that does not speaks anything.
22. table 1 – there is no phytase
23. table 1- there is no presented premix
24. it is unclear what table 2 relates to?
25. the same table 3
26. L 160- how there were measured?
27. 2.6. – it is unclear what parameters were calculated. Strength – it is general term
28. 2.7. L 163 these hormones are indicators of calcium and phosphorus metabolism. Bone metabolism is assessed on other indicators.
Markers of resorption represent degradation of type 1 collagen, include N-telopeptides, C-telopeptides, hydroxyproline, and the collagen crosslinks pyridinoline and deoxypyridinoline; acid phosphatase, a marker of osteoclast activity, and urinary calcium are also indicators of bone resorption. Bone formation markers indicate osteoblast activity; bone-specific alkaline phosphatase and the N-terminal and C-terminal extension peptides of procollagen reflect formation of organic matrix in bone. Osteocalcin, produced by osteoblasts but also released during osteoclastic degradation, may indicate either formation when resorption and formation are coupled or turnover when they are uncoupled.
29. L 174 - it is not mentioned that kidney were taken
30. L 176 – human cotransporter was assessed in birds? Predicted?
31. the statistical analysis is wrong
32. L 195 – from how many birds?
33. L 197 – eggshell strength? It is not informed in material and methods
34. L 196-198 - Basal internal egg quality is not presented in material and methods. It is unclear
35. L 213- how they were determined? It is unclear
36. L 215 – what strength? Unclear
37. L 220 – 1,25-(OH)2D3, calcitonin and parathormone are not indicators of bone metabolism
38. L 271- real activity was not detected
39. nothing is known how long the birds were given these diets.
Reviewer 3 Report
The paper contains valuable data. Results were properly reported, and the findings have been accurately discussed and compared with other published papers. For further improvement of the manuscript, it requires some modification.
P1,L14 = Simple Summary
Change “Dietary inorganic phosphorus input of laying hens should be tightly controlled in order to reduce feed cost and avoid environmental.” to be “In order to reduce feed costs and avoid environmental harm, laying hens should consume a limited amount of inorganic phosphorus in their diet.”.
P1,L24 = Abstract
Please avoid to use “We”, “I” in abstract, please change the sentence.
P2,L56 = Introduction
The introduction needs to be entirely re-written. It is very vague, and does not give the reader the necessary context to understand why you used the treatments you did, and why you made the measurements that you did. Please add new references to introduce your subject area, and compare it with other published material.
You can use new references such as:
Mokhtarzadeh, S., Nobakht, A., Mehmannavaz, Y., Palangi, V., Eseceli, H., & Lackner, M. (2022). Impacts of Continuous and Intermittent Use of Bovine Colostrum on Laying Japanese Quails: Egg Performance and Traits, Blood Biochemical and Antioxidant Status. Animals, 12(20), 2811.
Nobakht, A., Palangi, V., Ayaşan, T., & Coçkun, I. (2022). Efficacy of Tragopogon graminifolius medicinal powder as an inulin source for laying hens. South African Journal of Animal Science, 52(3), 252-258.
P2,L92 = Materials and Methods
Please revise the statistical analysis and include the experimental model.
P6,L192 = Results
Results explained adequately.
P11,L263 = Discussion
Discussion explained adequately.
P13,L364 = References
Some of the references are old, should be replaced.
Regards
Round 2
Reviewer 2 Report
Comments to the Authors of manuscript number: animals-2223882 entitled “Effects of Low-phosphorus Diets Supplemented with Phytase on the Production Performance and Bone Metabolism of Aged Hy-Line Brown Laying Hens”.
There is too much substantial methodological mistakes. For this reason the paper is not suitable for publication.
1. L. 17- environmental what? Is there like “environmental harm”?
2. The real activity of phytase should be given.
3. The composition of the diet is still unclear. There should be presented all diets because of it is not known how all diets were formulated. How all concentration were received if “Phytase and dicalcium phosphate were added to the 3% premix by pre-mixing step by step to ensure uniform mixing”.
“Phytase and dicalcium phosphate were added to the 3% premix by pre-mixing step by step to ensure uniform mixing. Farmers formulate diets by mixing corn, soybean meal, calcium carbonate, soybean oil, and palm meal with the 3% complex premix.”
Is it still 3%?
Author Response
Point 1:L. 17- environmental what? Is there like “environmental harm”? Response 1: "environmental harm" L15 Point 2: The real activity of phytase should be given. Response 2: Measured value: 50511 FTU/g; L123 Point 3: The composition of the diet is still unclear. There should be presented all diets because of it is not known how all diets were formulated. How all concentration were received if “Phytase and dicalcium phosphate were added to the 3% premix by pre-mixing step by step to ensure uniform mixing”. “Phytase and dicalcium phosphate were added to the 3% premix by pre-mixing step by step to ensure uniform mixing. Farmers formulate diets by mixing corn, soybean meal, calcium carbonate, soybean oil, and palm meal with the 3% complex premix.” Is it still 3% Response 3: The composition of the corn-soybean meal-based diet lacking inorganic phosphate is shown in Table 1. Briefly, the basal diet contained crude protein (15.79%), calcium (3.81%), NPP (0.12%), and phytase (1,470 FTU/kg). The premix provided the following per kilogram of diet: Vitamin A, 8100 IU; Vitamin D3, 2700 IU; Vitamin E, 24 mg; Vitamin K3, 3.0 mg; Vitamin B1, 2.4 mg; Vitamin B2, 6 mg; Vitamin B6, 3 mg; Niacin, 10 mg; Folic acid, 1.2 mg; Pantothenic, 7.8 mg; VB12, 0.003 mg; Copper (from CuSO4.5H2O), 10.5 mg; Manganese (from MnSO4.H2O), 84 mg; Zinc (from ZnSO4.6H2O), 78 mg; Iron (from FeSO4.6H2O), 54.0 mg; Selenium (from Na2SeO3.5H2O), 0.5 mg; Iodine (from KT), 0.45 mg; Lysine, 0.7g; Methionine, 1.77g; NaCl, 3g. All hens are fed the same basal diet. Dicalcium phosphate with different NPP levels was added in the experimental group. Phytase was used at 1470 FTU/kg in diets.
